# NHC-catalyzed enantioselective access to β-cyano carboxylic esters via in situ substrate alternation and release

Qingyun Wang[1,5], Shuquan Wu [2,5], Juan Zou[3,5], Xuyang Liang[1], Chengli Mou[3], Pengcheng Zheng [1] ✉ & Yonggui Robin Chi [1,4] ✉

A carbene-catalyzed asymmetric access to chiral β-cyano carboxylic esters is disclosed. The reaction proceeds between β,β-disubstituted enals and aromatic thiols involving enantioselective protonation of enal β-carbon. Two main factors contribute to the success of this reaction. One involves in situ ultrafast addition of the aromatic thiol substrates to the carbon-carbon double bond of the enal substrate. This reaction converts almost all enal substrate to a Thiol-click Intermediate, significantly reducing aromatic thiol substrates concentration and suppressing the homo-coupling reaction of enals. Another factor is an in situ release of enal substrate from the Thiol-click Intermediate for the desired reaction to proceed effectively. The optically enriched β-cyano carboxylic esters from our method can be readily transformed to medicines that include γ-aminobutyric acids derivatives such as Rolipram. In addition to synthetic utilities, our control of reaction outcomes via in situ substrate modulation and release can likely inspire future reaction development.

Cyano is a basic structural motif in bioactive molecules and synthetic building blocks[1–10]. A non-comprehensive survive of the literature indicates that more than fifty drug molecules contain one or multiple cyano groups, covering a variety of diseases such as cancers (Fig. 1a)[11–14]. For example, Cilomilast[15,16], a phosphodiesterase-4 (PDE4) inhibitor, is developed for the treatment of respiratory disorders. Deltamethrin[17,18] is a widely used insecticide with high efficacy among the pyrethroid insecticide families. The cyano unit is also a very convenient group for the synthesis of medicinal molecules such as non-natural amino acids (e.g. γ-aminobutyric acids (GABA) and their derivatives) for the treatment of neuro diseases that include Parkinson's disease and Huntington's disease[19–35]. Given the proven applications, efficient methods for the synthesis of cyano-containing molecules especially in enantioselective manners, continue to receive consideratinale attentions (Fig. 1b). Common synthetic methods include metal-catalyzed asymmetric carbon-carbon bond couplings[36–56] and

enantioselective hydrogenations[57–64]. Merits and limitations exist in these reported methods. For instance, highly toxic cyano salts were often used as the cyano sources in the metal-catalyzed coupling of alkene with cyano anion[43,47,48,64]. The use of flammable hydrogen gas for reductions may also bear limitations such as tolerance with other reducible functional groups especially under high pressure[61,62,64].

Here we report an approach for efficient and selective access to β-cyano carboxylic esters which can be easily converted into GABA derivatives with high optical purities (Fig. 1c). Our study was motivated by our objectives in constructing chiral cyano-containing molecules[65] and non-natural amino acid derivatives[66] via N-heterocyclic carbene (NHC) catalysis[67–92]. Our reaction starts with β-cyano enal (**1a**) as a key substrate. Under most reaction conditions, including those reported by Bode[93], Scheidt[94–98] and Huang[99–101] for β-protonation of other types of enal molecules, the substrate **1a** underwent homo-couplings to form **3a'** (Table 1)[59,93]. We then

[1]National Key Laboratory of Green Pesticide, Key Laboratory of Green Pesticide and Agricultural Bioengineering, Ministry of Education, Guizhou University, Guiyang 550025, China. [2]Center for Industrial Catalysis and Cleaning Process Development, School of Chemical Engineering, Guizhou Minzu University, Guiyang 550025, China. [3]School of Pharmacy, Guizhou University of Traditional Chinese Medicine, Guiyang 550025, China. [4]School of Chemistry, Chemical Engineering, and Biotechnology, Nanyang Technological University, Singapore 637371, Singapore. [5]These authors contributed equally: Qingyun Wang, Shuquan Wu, Juan Zou. ✉e-mail: zhengpc1986@163.com; robinchi@ntu.edu.sg

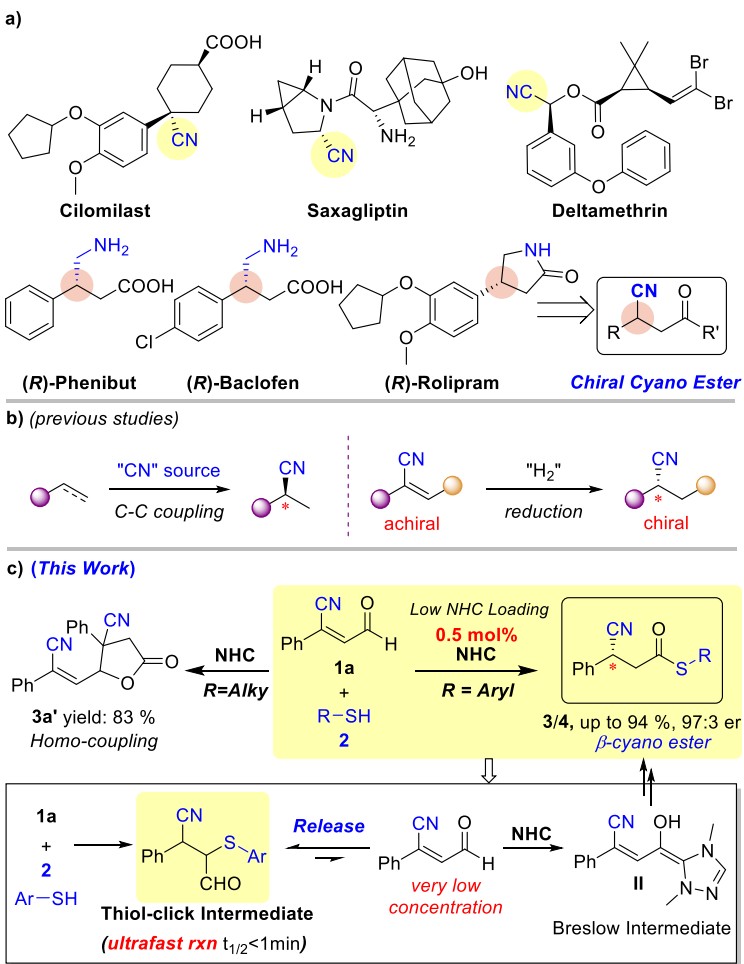

**Fig. 1 | Chiral cyano-containing functional molecules and the synthesis of chiral cyano group. a** Bioactive molecules bearing cyanos or prepared from cyano compounds. **b** Common method for synthesis of chiral cyano molecules. **c** NHC-catalyzed access to cyanos via modulated reaction pathway.

## Table 1 | Initial studies of nucleophiles[a]

| Entry | Nucleophile | Desired product (%)[b] 3a | Homo-coupling (%)[b] 3a' |
|---|---|---|---|
| 1 | MeOH | 0 | 82 |
| 2 | EtOH | 0 | 77 |
| 3 | PhOH | 0 | 81 |
| 4 | **2a** | 80 | 0 |
| 5 | **2a-1** to **2a-4** | 0 | 67-83 |
| 6 | **2a-5** to **2a-6** | 0 | 0 |

[a]Unless otherwise specified, the reactions were conducted with **1a** (0.10 mmol), nucleophiles (0.10 mmol), pre-NHC **A** (0.01 mmol), base (0.02 mmol) and solvents (2.0 mL) at rt for 12 hrs. [b]Isolated yield of **3a** and **3a'**.

reasoned that the concentration of enal **1a** must be dramatically reduced (Fig. 1c). A survey of nucleophiles that can undergo facile 1,4-addition with enal **1a** revealed that aromatic thiols can quickly react with enals almost quantitatively. Under such a condition with enal nearly undetectable, homo-coupling product (**3a'**) was completely suppressed. At the same time, the desired β-cyano carboxylic ester products **3** and **4** can be obtained with excellent yields and er values. The dynamic Thiol-Michael addition click reaction[102–109] of enal and thiol (and release of enal substrate from the Thiol-Michael click adduct) provide a good control over the reaction pathways and outcomes. Our reaction can be easily scale up (open to five grams) with as little as 0.5 mol% NHC catalyst. The β-cyano carboxylic ester products from our reactions can be quickly transferred to many non-natural amino acid-based pharmaceuti-cals such as Phenibut, Baclofen and Rolipram[19,23,26,32]. From the reaction design point of view, our approach via in situ substrate alteration and release for reaction controls may provide solutions for reaction discoveries and practical synthesis.

## Results

### Reaction development
Our study starts with β-cyano enal as the substrate (Table 1). Inspired by studies from previous works[93–101], the alcohol and phenol were chosen as the nucleophiles and proton sources (Table 1, entries 1 to 3). Then the benzyl mercaptan, aliphatic thiols and amines were used as nucleophiles (Table 1, entry 5 to 6). Unfortunately, under those conditions, only homo-coupling of enal (to afford **3a'**) was observed. When aromatic thiols (such as p-toluene-thiol, **2a**) were used (Table 1, entry 4), the desired β-cyano carboxylic ester **3a** was obtained with 80% yield.

Then, we employed β-cyano enal and p-toluene-thiol as the model reaction substrates to search for optimal conditions under various NHC catalysts (Table 2). The desired β-cyano carboxylic ester (**3a**) was disclosed in 74% isolated yield with potential enantioselectivity when $K_2CO_3$ was used as base in the presence of THF under aminoindanol-derived thiazolium pre-NHC **A** (Table 2, entry 1). The target product was achieved slightly lower yield and er value when the N-Phenyl substituent on NHC catalyst was replaced with N-Mesityl group (**B**) (Table 2, entry 2). The chiral benzyl substituted morpholine-based pre-NHC **C** gave the lower er value and yield compared with other catalysts (Table 2, entry 3). Likewise, the good product yield but poor enantioselectivity observed when switched to catalyst **D** (Table 2, entry 4). We then evaluated the effect of pre-NHC **A** for the reaction system, found that the inorganic base $Cs_2CO_3$ significantly decreased the product enantioselectivity although high yield was obtained (Table 2, entry 5). To our surprise, the reaction efficiency was notably improved when organic bases such as DMAP, $Et_3N$ and DABCO were used, and the results showed that DABCO could be the most suitable base to further optimize the reaction condition (Table 2, entries 6 to 8). The effect of solvent was also examined, and toluene exhibited to be the suitable solvent (Table 2, entries 9 to 12). Finally, the optimal reaction result was afforded when 4 Å MS was chosen as the additive, the corresponding chiral β-cyano carboxylic ester was provided in 83% isolated yield with excellent enantioselectivity (95:5 er) (Table 2, entry 13). The absolute configuration of **3a** was confirmed by X-ray crystallographic analysis.

### Substrate scope
Having established the optimal reaction condition for this NHC-catalyzed hydro-thioesterification, the examples of the reaction was

## Table 2 | Condition optimization[a]

| Entry | Pre-NHC | Base | Solvent | Yield (%)[b] | Er[c] |
|---|---|---|---|---|---|
| 1 | **A** | $K_2CO_3$ | THF | 74 | 84:16 |
| 2 | **B** | $K_2CO_3$ | THF | 64 | 62:38 |
| 3 | **C** | $K_2CO_3$ | THF | 71 | 64:36 |
| 4 | **D** | $K_2CO_3$ | THF | 85 | 55:45 |
| 5 | **A** | $Cs_2CO_3$ | THF | 83 | 50:50 |
| 6 | **A** | DMAP | THF | 76 | 88:12 |
| 7 | **A** | $Et_3N$ | THF | 62 | 71:29 |
| 8 | **A** | DABCO | THF | 88 | 85:15 |
| 9 | **A** | DABCO | DCM | 73 | 89:11 |
| 10 | **A** | DABCO | EtOAc | 78 | 94:6 |
| 11 | **A** | DABCO | MTBE | 56 | 91:9 |
| 12 | **A** | DABCO | Toluene | 81 | 94:6 |
| 13[d] | **A** | DABCO | Toluene | 83 | 95:5 |

[a]Unless otherwise specified, the reactions were conducted with **1a** (0.10 mmol), **2a** (0.12 mmol), pre-NHCs (0.01 mmol), bases (0.02 mmol) and solvents (2.0 mL) at 30 °C for 11 hrs.
[b]Isolated yield of **3a**.
[c]The er values of **3a** were determined via HPLC on the chiral stationary phase.
[d]100 mg 4Å MS was used.

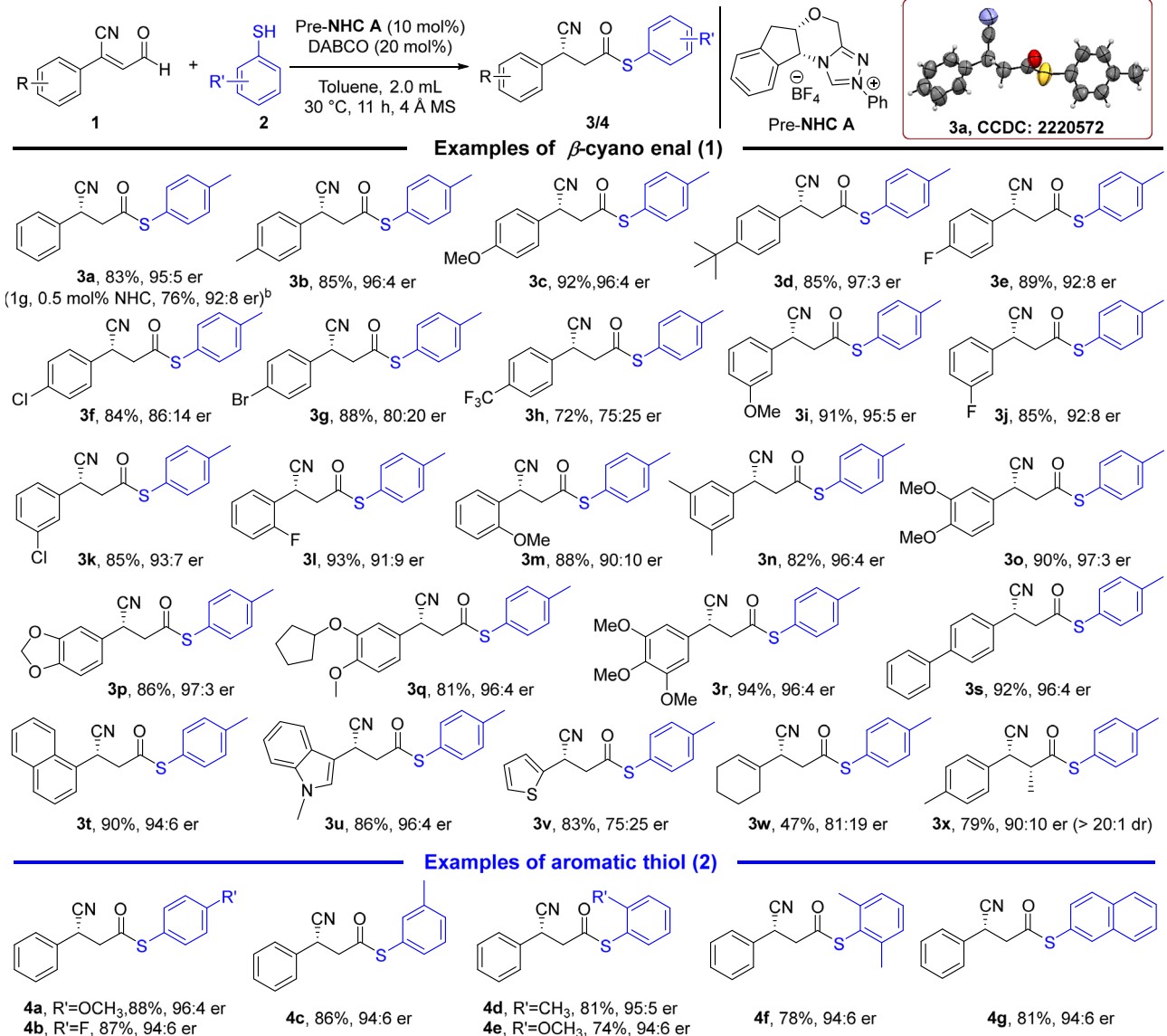

**Fig. 2 | Substrate scope[a].** [a]Reaction conditions as stated in Table 2, entry 13, yields were isolated yields after purification by column chromatography, er values were determined via HPLC on chiral stationary phase. [b]The reaction was carried out at 1 gram-scale based on **1a** (6.4 mmol), 0.0005 mmol pre-NHC **A**, reaction time was 18 hrs.

examined with regard to the β-cyano enal substrates (Fig. 2). Various substituents installed on the phenyl of β-cyano enal were tolerated in this reaction condition. The good yields and excellent enantioselectivities (up to 97:3 er) were observed when para- position of the phenyl ring was substituted with electron-donating groups such as methoxyl, methyl and t-butyl group (**3b** to **3d**). Slightly lower enantioselectivity was observed when electron-deficient fluoro-atom appeared on the para- position compared with model reaction (**3e**). The similar results showed when the substituents of phenyl ring bearing other electron-withdrawing groups such as chrolo- and bromo-atoms (**3f** to **3h**). Similarly, the substituents at the meta- position of the benzene ring also showed the same reaction regularities (**3i** to **3k**). However, the reaction results of the ortho-substituents were different, the er values of both electron-withdrawing and electron-donating groups decreased slightly (**3l** to **3m**). It suggested that possibly caused by the unfavourable steric hindrance on the position. Other muti-substituents were also suitable for the construction of reaction system, such as dimethyl, dimethoxy and piperonyl, they all afforded excellent yields and excellent enantioselectivities (**3n** to **3p**). Notably, the medicinally valuable product **3q** had excellent yield and excellent er value. Meanwhile, high yield and excellent er value was also obtained for the

phenyl ring bearing muti-substituent like trimethoxy (**3r**). Replacement with biphenyl and napthyl group afforded in more than 90% yields with excellent enantioselectivities (**3s** and **3t**). Heterocyclic and alkyl group such as thiophene, indole and cyclohexene were also used in these reactions, obtained excellent to acceptable yields and enantioselectivities (**3u** to **3w**). Meanwhile, the α-methyl substituted β-cyano enal was well-tolerated, the desired product **3x** was produced in good yield and er value. Subsequently, reducing the catalyst loading to 0.5 mol% (pre-NHC **A**) and the model reaction was conducted on a gram scale, **3a** can be obtained in 76% yield with good enantioselectivity.

The examples of the substituent aromatic thiols also be examined, various of substituted aromatic thiols such as meta-, ortho-methyl phenylthiol, steric dimethyl aromatic thiols, methoxyl phenylthiol and napthalene group can be matched the reactions, give the good to excellent yields and excellent enantioselectivities (**4a** to **4g**).

**Synthetic transformations**
Furthermore, in order to demonstrate the synthetic utility of this methodology, large-scale experiment was achieved as shown in Fig. 3a. The catalyst loading can be reduced to 1 mol% and the reaction was

**Fig. 3 | Synthetic transformations to GABA derivatives. a** Potenial industrial application reaction with 1 mol% NHC loading. **b** Synthesis of other chiral GABA derivatives: (R)-Phenibut and (R)-Baclofen.

conducted on 5 gram scale (190 times scale up), **3q** can be obtained with good yield (71%) and excellent enantioselectivity (95:5 er), it suggested that our methodology had the potential industrial application prospects. Then, (R)-Rolipram **6** can be synthesized by an efficient synthetic protocol. The product **3q** was transesterified with MeOH under the catalysis of sulfuric acid to provide the corresponding methyl ester **5** in high yield, which is a biologically active molecule and important building block that can be easily converted into GABA-derivatives. Then, **5** was reduced by $NiCl_2$/$NaBH_4$ in MeOH and the corresponding (R)-Rolipram **6** can be readily obtained in high yield with preserved enantioselectivity[63,110,111]. Meanwhile, the compounds **3a** and **3f** can be efficiently converted into the GABA drugs (R)-Phenibut and (R)-Baclofen (Fig. 3b) by using the same methods[110,111].

## Mechanistic studies

In the previous works[93–101], many examples of β-protonation reactions had been developed, but the in-depth mechanisms were under-developed and worth further exploring. We performed multiple studies to investigate the reaction mechanism. In the initial studies, it was found that the reaction outcomes can be modulated through different substrates (Table 1). The gas chromatography mass spectrometry (GC-MS) was employed to investigate the relationship between substrate concentration and reaction time (Fig. 4a). First, the concentration of **1a** (model reaction, Table 2, entry 13) was detected by GC-MS and it was found that substrate **1a** was rapidly consumed ($t_{1/2} < 1$ min), but no corresponding product **3a** can be detected within 4 min, which means that the consumption of **1a** was not synchronized with the formation of **3a**. Furthermore, as a comparative experiment, **2a-1** was used to study the mechanism of modulating reaction outcomes (Table 1, entry 5). It was found that the consumption rate of **1a** was much slower than the model reaction (Table 2, entry 13) when **2a-1** existed. Meanwhile, the concentration of **2a-1** had remained during the whole reaction process. The results suggested that the reaction outcomes can be modulated by different nucleophiles, such as **2a** and **2a-1**.

Based on the results of GC-MS, it was suggested that some intermediate was generated very fast from the enal substrate **1a**, then the intermediate can be gradually converted into desired product **3a**. Fortunately, a new intermediate can be clearly detected after two minutes by TLC in the model reaction. Meanwhile, this new intermediate (**I**) can be synthesized independently by DABCO catalysis without NHC (Fig. 4b), and isolated after 15 min with 91% yield. The structure of intermediate was confirmed by X-ray diffraction (**I′**). Subsequently, the Thiol-click Intermediate **I** was used as a substrate to react with NHC (model reaction condition), the desired product **3a** can be obtained. The $pK_a$ of the thiols along with their structures were important factors that impact the Thiol-Michael addition click reaction. The ultrafast Thiol-Michael addition click reaction was more likely to occur when the $pK_a$ value of substrates **2** are less than 10 (aromatic thiols, $pK_a = 7-8$, aliphatic thiols, $pK_a = 10-11$)[102,109]. Thus, the formation of Thiol-click Intermediate **I** from β-cyano enal and aromatic thiol can extremely decrease the concentration of **1a**. Therefore, the side reaction pathway had been inhibited, the homo-coupling product **3a′** cannot be detected in the reaction.

To explore the formation of **3a** from Thiol-click Intermediate **I**, the liquid chromatography-high resolution mass spectrum (LC-HRMS) was employed to detect the key intermediates in the model reaction (Fig. 5a)[112–115]. There were two possible reaction pathways (NHC reacts with substrate **1a** and NHC reacts with Thiol-click Intermediate **I**) need to be confirmed. The Breslow intermediate and the acylazolium intermediate are isomers, it was difficult to directly identify these two intermediates by mass-to-charge ratio. Therefore, additional experiment was performed to identify the retention time of the Breslow intermediate. In the previous work, the β-protonation reaction can be inhibited when the strong base existed in the reaction[93,116,117]. Therefore, DBU was chosen as the strong base to assist the generation of the Breslow intermediate **II** (Entry A), and the Breslow intermediate **II** can be detected (Rt = 5.91) by LC-HRMS. In the same LC-HRMS condition, two peaks ($m/z = 447.1816$) were found (4.85 min and 5.98 min) in

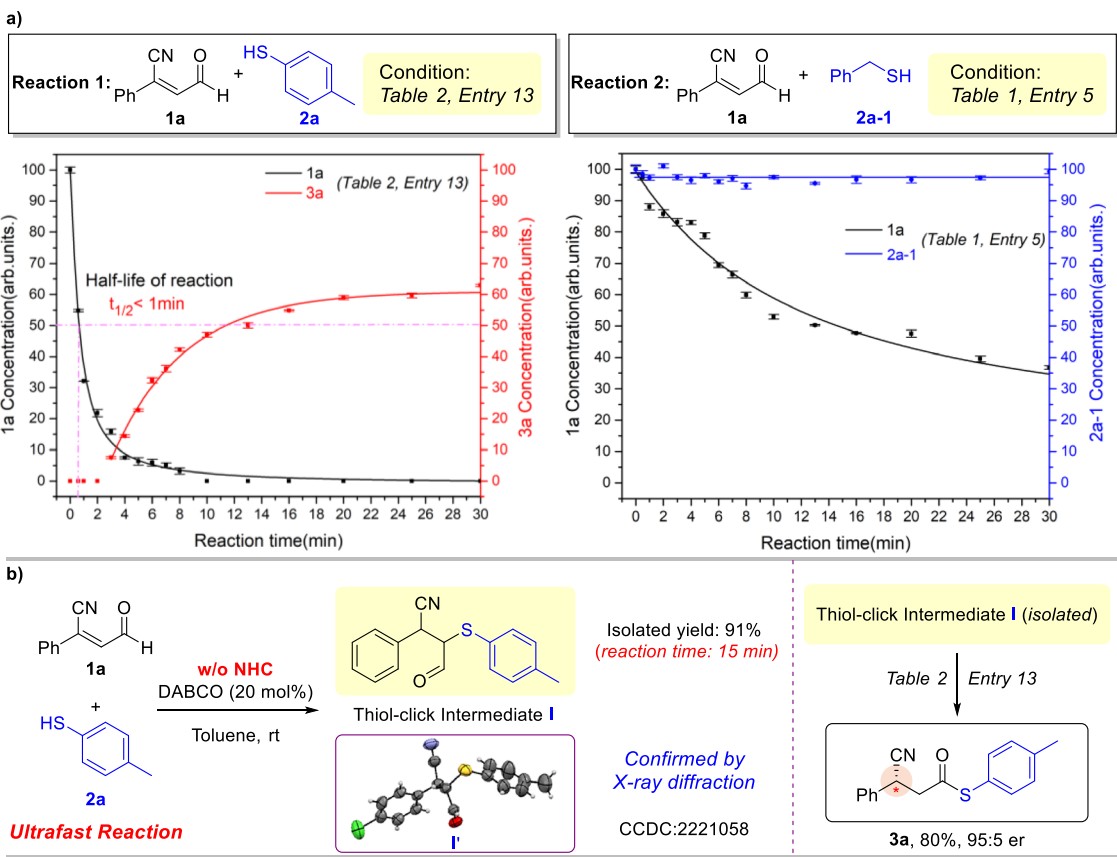

**Fig. 4 | The research of Thiol-click intermediate. a** Substrate concentration monitored via GC-MS. **b** Synthesis and confirmation of Thiol-click Intermediate.

Entry B, compared with the Entry A, the peak at 5.98 min can be identified as the Breslow intermediate **II**, and another peak (Rt = 4.85) was attributed to the acylazolium intermediate **III**.

Further investigated the data of LC-HRMS, a semiquantitative analysis was performed. The intensity of the Breslow intermediate **II** was obviously higher than that of the acylazolium intermediate **III**. Meanwhile, a trace acylazolium intermediate of NHC and Thiol-click Intermediate **I** (see Supplementary Information) was detected in the reaction, which means the NHC can react with Thiol-click Intermediate **I**, but the acylazolium intermediate was still bearing the thiol moiety after oxidation. Therefore, the pathway by which the NHC reacted with Thiol-click Intermediate **I** cannot obtain the desired product **3a**.

According to the research of key intermediates, the NHC was preferentially reacted with substrate **1a** to form Breslow intermediate **II** rather than the Thiol-click Intermediate **I**. Then, the acylazolium intermediate **III** was formed from the Breslow intermediate **II** rather than the redox reaction of Thiol-click Intermediate **I**. Thus, it was suggested that the reactivity of β-cyano enal substrate **1a**, an enal which was bearing a strong electron-withdrawing group, was higher than the Thiol-click Intermediate **I**. The NHC was preferentially reacted with **1a** to form the corresponding Breslow intermediate.

Based on these mechanism studies, the possible mechanism was proposed (Fig. 5b). The Thiol-click Intermediate **I** was generated from substrates **1a** and **2a** through an ultrafast Thiol-Michael addition click reaction, via in situ substrate alternation, which was catalyzed by DABCO. During this reversible reaction process, **1a** can be facile released in a very low concentration from the Thiol-click Intermediate **I**. Then **1a** was added with NHC to form the corresponding Breslow intermediate **II**. Subsequently, the acylazolium intermediate **III** was obtained from the Breslow intermediate **II** by asymmetric protonation. Then, the substrate **2a** was reacted with the acylazolium intermediate

**III** to form the desired product **3a**. Without the in situ substrate alternation, high concentration of **1a** will lead to homo-coupling reaction. It means that the reaction outcomes were related to the concentration of substrates.

In summary, we developed an organic catalytic access to chiral β-cyano carboxylic esters. Aromatic thiols were used to react with β,β-disubstituted enals to form the corresponding desired products involving enal β-carbon protonation as an enantio-determining step. Keys to the success of our approaches include an ultrafast Thiol-Michael addition click reaction between enals and aromatic thiols that dramatically reduced the concentration and inhibits undesired homo-coupling of the enal substrates. A facile reversed reaction of the Thiol-Michael click adduct effectively releases enal substrate for the desired reaction to proceed to form the β-cyano carboxylic ester products. Our strategy in controlling reaction pathways via in situ substrate modulation can be further used in developing new reactions especially those where effective concentration of the substates matter. The desired β-cyano carboxylic ester products from our reactions, easily obtained in scalable operations with low catalyst loadings, can be readily converted to GABA medicines such as Rolipram, Phenibut and Baclofen.

## Methods
### General procedure for the catalytic reactions
To a dry 4.0 mL vial equipped with a magnetic stir bar, **1** (0.10 mmol), **2** (0.12 mmol), pre-NHC **A** (0.01 mmol) and DABCO (0.02 mmol) were added. After purges with $N_2$ in glove-box, anhydrous Toluene (2.0 mL), and 4 Å MS (100 mg) was added and sealed. The reaction mixture was stirred at 30 °C for 11 hrs. Then the mixture was directly concentrated under reduced pressure to afford a crude product. The crude product was purified via column chromatography on silica gel (petroleum ether/ethyl acetate = 15/1) to afford the desired product **3/4**.

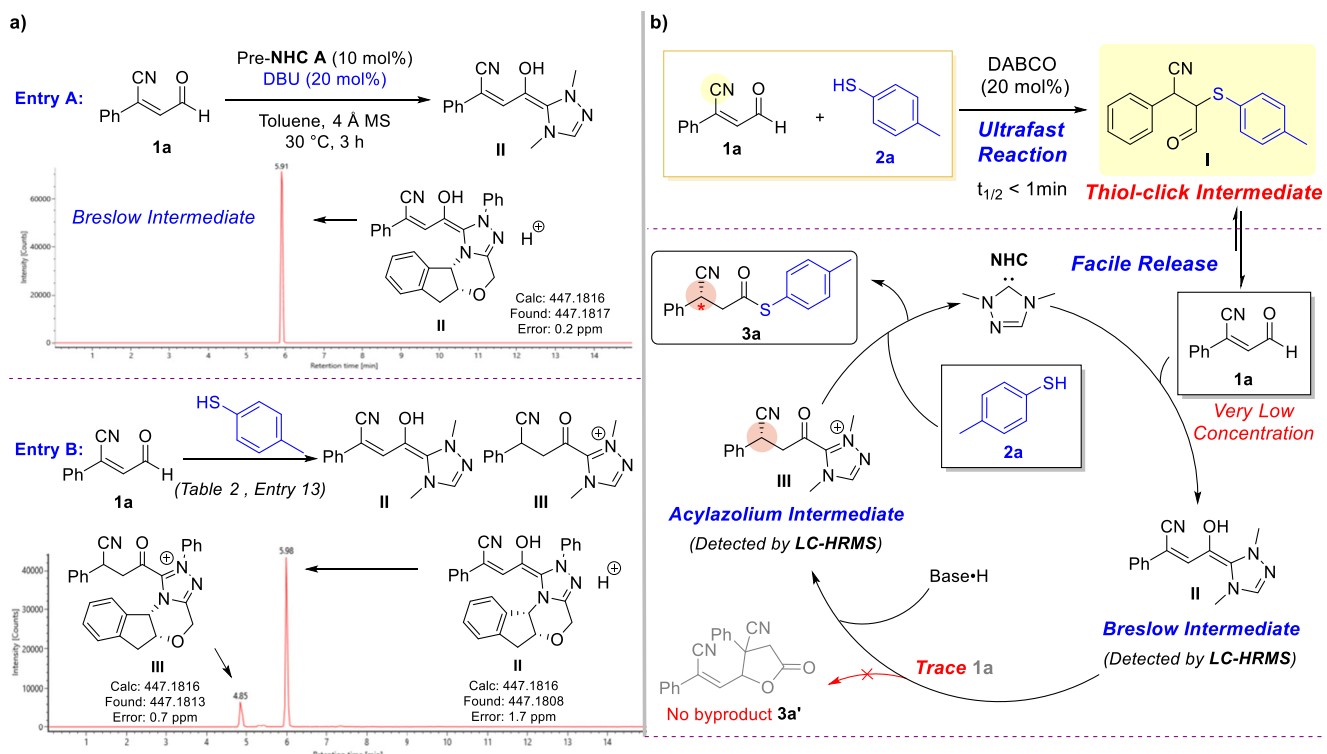

**Fig. 5 | LC-HRMS results and proposed catalytic reaction pathway. a** Key intermediate of model reaction confirmed via LC-HRMS. **b** Proposed pathway: in situ substrate alternation and release strategy.

## General procedure for the scale-up catalytic reactions

To a 100.0 mL over-dried round bottom flask equipped with a magnetic stir bar, **1a** (6.40 mmol, 1 g), **2a** (7.68 mmol, 0.95 g), pre-NHC **A** (0.0005 mmol) and DABCO (0.001 mmol) were added. The flask was then sealed, purged and backfilled with N₂ three times in glovebox before adding Toluene (60.0 mL). The reaction mixture was stirred at 30 °C for 18 hrs. Then the mixture was directly concentrated under reduced pressure to afford a crude product. The crude product was purified via column chromatography on silica gel (petroleum ether/ethyl acetate = 15/1) to afford the desired product **3a** in 76% yield and 92:8 er.

## Data availability

The experimental method and data generated in this study are provided in the Supplementary Information file. The crystallographic data for structures of **1k**, **3a**, **3x** and **I′** have been deposited in the Cambridge Crystallographic Data Centre under accession CCDC code 2220606, 2220572, 2277728 and 2221058, respectively. Copies of the data can be obtained free of charge via www.ccdc.cam.ac.uk/data_request/cif. All other data are available from the authors upon request.

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

## Acknowledgements

We acknowledge funding supports from the National Natural Science Foundation of China (21732002, 22061007, P.C.Z. and 22071036, Y.R.C.); Frontiers Science Center for Asymmetric Synthesis and Medicinal Molecules, Department of Education, Guizhou Province [Qianjiaohe KY (2020)004, Y.R.C.]; The 10 Talent Plan (Shicengci) of Guizhou Province ([2016] 5649, Y.R.C.); Science and Technology Department of Guizhou Province [Qiankehe-jichu-ZK[2022]zhongdian024, P.C.Z.], ([2018]2802, [2019]1020, Y.R.C.); Program of Introducing Talents of Discipline to Universities of China (111 Program, D20023, Y.R.C.) at Guizhou University; Singapore National Research Foundation under its NRF Investigatorship (NRF-NRFI2016-06, Y.R.C.) and Competitive Research Program (NRF-CRP22-2019-0002, Y.R.C.); Ministry of Education, Singapore, under its MOE AcRF Tier 1 Award (RG7/20, RG5/19, Y.R.C.), MOE AcRF Tier 2 (MOE2019-T2-2-117, Y.R.C.), and MOE AcRF Tier 3 Award (MOE2018-T3-1-003, Y.R.C.); a Nanyang Research Award Grant; and a Chair Professorship Grant, Nanyang Technological University.

## Author contributions

Q.Y.W. and J.Z. conducted most of the experiments. S.Q.W. contributed to designs. X.Y.L. and C.L.M. conducted some experiments. P.C.Z. and Y.R.C. conceptualized and directed the project and drafted the manuscript with assistance from all coauthors. All authors contributed to part of the experiments and/or discussions. Q.Y.W. and S.Q.W. and J.Z. are estimated to contribute equally to this work.

## Competing interests

The authors declare no competing interests.
