## [Peer Review File · Nature Communications]

NHC-Catalyzed Enantioselective Access to β -Cyano Carboxylic Esters via In Situ Substrate Alternation and ReleaseReviewers' Comments:

Reviewer #1:

Remarks to the Author:

This manuscript describes N-heterocyclic carbene-catalyzed asymmetric access to chiral β -nitrile carboxylic esters. Aromatic thiols and β -nitrile enals were utilized as the substrates with protonation of the enal β -carbon atom as the enantio-determining step. The Thiol-Michael addition between enals and aromatic thiols was the key to the success of this approach, which perfectly controlled the reaction pathway to form the desired products and inhibit the common homocoupling products of enals. Although the reaction is applicable to limited substrates, especially the nucleophile substrates, this method still could inspire the readers, especially in the aspect of in situ substrate modulation to control the reaction pathways. The publication of this work in Nat. Commun. is recommended after authors addressing the following concerns and questions.

1. The substrate scope seems limited. Would β -alkyl- β -nitrile enals be applicable for this reaction? What about other nucleophiles such as amines instead of thiols? It's recommended to provide more attempts and details on the substrate scope.
2. There are some errors and typos need to be corrected.
 - 1) Page 2, Table 2, footnote d should mark in entry 13.
 - 2) Page 2, Line 132, "Various of substituents..." should be "Various substituents..."
 - 3) Page 5, Line 199, 200 and 201, the character K of pKa should be italic.
 - 4) Page 5, Line 220, "According to the research of key intermediates. The ..." should be "According to the research of key intermediates, the ..."
 - 5) Page 6, Line 249, "bar, was add d 1a" should be "bar, was added 1a".

Reviewer #2:

Remarks to the Author:

Asymmetric synthesis of nitriles has attracted substantial attention among organic community. The traditional methods involve the metal-catalyzed coupling of alkenes with cyano sources and the catalytic hydrogenation of prochiral nitriles. Herein, Chi, Zheng and co-workers reported a new strategy for the enantioselective synthesis of chiral nitrile esters by NHC-catalyzed beta-protonation of prochiral nitrile enals. The reaction involved an interesting thiol-click intermediate which was determined to be the key success of this reaction. The substrate scope, synthetic application and reaction mechanism were well investigated and demonstrated. Basically, this work merits publication in Nat. Commun., but some issues need to be addressed.

1. As for naming the title compounds, is "beta-cyano carboxylic esters" more suitable than "beta-nitrile carboxylic esters"?
2. A recent asymmetric synthesis of nitriles via cooperative organocatalysis and PC catalysis needs to be included in introduction section (Nature Catalysis, 2023, doi.org/10.1038/s41929-023-00939-y). More reviews of NHC-organocatalysis need to be included.
3. How to explain that the er values of 3f-h decreased significantly?
4. Examples of alpha-substituted enals need to be tested to further expand the utility of this protocol.
5. A control experiment of the conversion of Intermediate I to product 3a under NHC catalysis needs to be included.
6. Please adjust the figures in Page S84.

Enclosed is our revised manuscript " **NHC-Catalyzed Enantioselective Access to β -Nitrile Carboxylic Esters via In Situ Substrate Alternation and Release** "

We are pleased by the supporting comments from all the reviewers. A point-by-point response to the editorial requests and referee comments is included in this letter.

- Page **2-3**: response to **Reviewer #1**
- Page **4-7**: response to **Reviewer #2**

Response to Reviewer #1

Reviewer #1 has recommended publication after addressing the following concerns and questions.

1). The substrate scope seems limited. Would β -alkyl- β -nitrile enals be applicable for this reaction? What about other nucleophiles such as amines instead of thiols? it's recommended to provide more attempts and details on the substrate scope.

Our Response:

1-1. The referee is right it will be valuable for this chemistry to work with β -alkyl substrates. Unfortunately, at this point we face technical difficulties to prepare such β -alkyl enals. We have tried a few common synthetic routes to prepare such substrate, but without success. A few of our attempted routes are shown below. We'll look into this matter again when there is a need in the future.

1-2. Under the reaction condition in manuscript Table 1 (Page 2, line 67), we attempted other nucleophiles to replace thiols such as amines to react with **1a**. The results were listed in Table R1.

Table R1. Initial studies of nucleophiles^a.

Entry	Nucleophile	Desired product(%) ^b	Homo-coupling(%) ^b 3a'
1	MeOH	0	82
2	EtOH	0	77
3	PhOH	0	81
4	2a	80	0
5	2a-1 to 2a-4	0	67-83
6	2a-5 to 2a-6	0	0

^aUnless otherwise specified, the reactions were conducted with **1a** (0.10 mmol), **nucleophiles** (0.10 mmol), *pre*-NHC **A** (0.01 mmol), base (0.02 mmol) and solvents (2.0 mL) at rt for 12 h. ^bIsolated yield of **3a** and **3a'**.

2). There are some errors and typos need to be corrected.

1. Page 2, Table 2, footnote d should mark in entry 13.
2. Page 2, Line 132, "Various of substituents..." should be "Various substituents..."
3. Page 5, Line 199, 200 and 201, the character K of pKa should be italic.
4. Page 5, Line 220, "According to the research of key intermediates. The ..." should be "According to the research of key intermediates, the ..."
5. Page 6, Line 249, "bar, was add d 1a" should be "bar, was added 1a".

Our Response: We have revised the above comments in the revised manuscript.

Response to Reviewer #2

Reviewer #2 has recommended publication, but some issues need to be addressed.

1). As for naming the title compounds, is “beta-cyano carboxylic esters” more suitable than “beta-nitrile carboxylic esters”?

Our Response: We have revised the comment in the revised manuscript.

2). A recent asymmetric synthesis of nitriles via cooperative organocatalysis and PC catalysis needs to be included in introduction section (Nature Catalysis, 2023, doi.org/10.1038/s41929-023-00939-y). More reviews of NHC-organocatalysis need to be included.

Our Response: Revised.

3). How to explain that the er values of 3f-h decreased significantly?

Our Response: Products containing electron withdrawing group such as chloro- and bromo-atoms (**3f-3h**), the β -CH is more acidic which will lead to racemization of the products. Therefore, several controlled experiments were carried out, er value of product **3f** in different reaction time were monitored. It was found that the er value decreased gradually with the extension of reaction time.

Table R2. The er value of **3f** in different reaction times

Entry	Reaction time / h	Er of 3f
1	12	84:16
2	18	81:19
3	30	77:23

	RT	Area	% Area	Height	% Height
1	38.622	2141753	50.31	31017	62.87
2	50.137	2115675	49.69	18314	37.13

	RT	Area	% Area	Height	% Height
1	39.217	8037841	83.85	124420	88.94
2	51.155	1548344	16.15	15478	11.06

	RT	Area	% Area	Height	% Height
1	38.010	5382084	80.96	78097	83.45
2	49.400	1265974	19.04	15492	16.55

	RT	Area	% Area	Height	% Height
1	39.712	4071935	77.24	45486	83.92
2	52.356	1199733	22.76	8716	16.08

4). Examples of α -substituted enals need to be tested to further expand the utility of this protocol.

Our Response: An α -methyl substituted substrate **1x** was synthesized. The corresponding product **3x** can be obtained under the model reaction condition with 79% yield and 90:10 er (> 20:1 dr). The structure was confirmed by X-ray crystallographic analysis.

Synthesis of **1x**

Catalytic Reaction

5). A control experiment of the conversion of Intermediate I to product **3a** under NHC catalysis needs be included.

Our Response: We have updated this control experiment in revised manuscript (Fig.3). The Thiol-click Intermediate I was used to directly react with NHC under model reaction condition, the desired product **3a** can be obtained with 80 % yield and 95:5 er.

6). Please adjust the figures in Page S84.

Our Response: Revised.

Reviewers' Comments:

Reviewer #1:

Remarks to the Author:

The revision of the paper titled "NHC-Catalyzed Enantioselective Access to β -Nitrile Carboxylic Esters via In Situ Substrate Alternation and Release" pertinently addresses the reviewers' previous comments. Screening data of other nucleophiles, the example of α -substituted enals, and the control experiment of the conversion of Intermediate I to product 3a under NHC catalysis were all supplemented in the revised manuscript and SI. The revisions made the description more accurate and conclusion stronger thus improve the scientific value of the study. Given those improvements, I would suggest the publication of this manuscript in Nat. Commun.

Reviewer #2:

Remarks to the Author:

The authors have well responded to the concerns raised by the referees. The current form can be accepted for publication.

Enclosed is our revised manuscript " **NHC-Catalyzed Enantioselective Access to β -Cyano Carboxylic Esters via In Situ Substrate Alternation and Release**".

We are pleased by the supporting comments from all the reviewers. A point-by-point response to the editorial requests and referee comments is included in this letter.

Reviewer #1 (Remarks to the Author):

The revision of the paper titled "NHC-Catalyzed Enantioselective Access to β -Nitrile Carboxylic Esters via In Situ Substrate Alternation and Release" pertinently addresses the reviewers' previous comments. Screening data of other nucleophiles, the example of α -substituted enals, and the control experiment of the conversion of Intermediate I to product 3a under NHC catalysis were all supplemented in the revised manuscript and SI. The revisions made the description more accurate and conclusion stronger thus improve the scientific value of the study. Given those improvements, I would suggest the publication of this manuscript in Nat. Commun.

Our Response: We are pleased that the manuscript has been accepted by the reviewer.

Reviewer #2 (Remarks to the Author):

The authors have well responded to the concerns raised by the referees. The current form can be accepted for publication.

Our Response: We are pleased that the manuscript has been accepted by the reviewer.